# Synthesis of Carbazole–Thiazole Dyes via One-Pot Tricomponent Reaction: Exploring Photophysical Properties, Tyrosinase Inhibition, and Molecular Docking

**DOI:** 10.3390/s24196368

**Published:** 2024-09-30

**Authors:** Przemysław Krawczyk, Beata Jędrzejewska, Joanna Cytarska, Klaudia Seklecka, Krzysztof Z. Łączkowski

**Affiliations:** 1Department of Physical Chemistry, Faculty of Pharmacy, Collegium Medicum, Nicolaus Copernicus University, Kurpińskiego 5, 85-950 Bydgoszcz, Poland; 2Faculty of Chemical Technology and Engineering, Bydgoszcz University of Science and Technology, Seminaryjna 3, 85-326 Bydgoszcz, Poland; beata@pbs.edu.pl; 3Department of Chemical Technology and Pharmaceuticals, Faculty of Pharmacy, Collegium Medicum, Nicolaus Copernicus University, Jurasza 2, 85-089 Bydgoszcz, Poland; cytar@cm.umk.pl (J.C.); krzysztof.laczkowski@cm.umk.pl (K.Z.Ł.)

**Keywords:** carbazole derivatives, optical functional materials, linear optical properties, NLO properties, tyrosinase, toxicology, bioimaging, protein affinity, TD-DFT method

## Abstract

Carbazole is an aromatic heterocyclic organic compound consisting of two fused benzene rings and a pyrrole ring and is a very valuable building structure for the design of many compounds for use in various fields of chemistry and medicine. This study presents three new carbazole-based thiazole derivatives that differ in the presence of a different halogen atom: chlorine, bromine, and fluorine. Experimental studies and quantum-chemical simulations show the effect of changing a halogen atom on the physicochemical, biological, and linear and nonlinear optical properties. We have also found that carbazoles C-Cl, C-Br, and C-F exhibit high tyrosinase inhibitory activity, with IC_50_ values in the range of 68–105 µM with mixed mechanism of action. Finally, molecular docking to the active site of Concanavalin A (ConA) and bioavailability for all compounds were evaluated.

## 1. Introduction

In recent years, increasing interest in metal-free organic materials has been observed due to their low production cost, unique optoelectronic properties, and the possibility of their use in optoelectronic devices, especially in solar cells. An additional advantage of metal-free organic materials is the ability to modulate their chemical structure to optimize absorption, fluorescence emission, and electrochemical properties for achieving better efficiency [1,2]. Many groups of compounds containing fluorene [3], triphenylamine [4], coumarin [5], indoline [6], and BODIPY [7] moieties have been studied so far to obtain compounds with the required electronic and optical properties, especially D-π-A types of dyes. Carbazole, a well-known heterocyclic aromatic system has been recently successfully employed in optoelectronics [8]. The carbazole ring is an electron-rich system characterized by high chemical and thermal stability, good absorption and emission properties, and due to its photoluminescence and hole-transport property, it has been successfully used in the production of photovoltaic systems, OLEDs [9], and DSSCs [10,11]. Carbazole-based polymers (PCz) are characterized by a higher redox potential and strong UV absorption compared to other polymers [12]. Unlike the electron-donating carbazole ring, thiazole is a widely used electron-accepting heterocyclic system. The properties of this system have led to the discovery of the novel thiazole fluorophores with excellent photophysical properties [13,14,15,16]. Tyrosinase is a Cu-containing oxidoreductase that controls melanin production in animals. Melanin is responsible for the color of the skin as well as its aesthetic features, such as freckles and age spots. Melanoma is the most invasive skin cancer which is characterized by a high mortality rate. The increased tyrosinase activity in these types of cancer leads to accumulation of melanin content, which protects cancer cells from chemo- and radiotherapy due to interaction with drugs and absorbing radiation. It is also responsible for the browning of damaged fruits or plant tissue. Therefore, we are currently looking for effective tyrosinase inhibitors that can be used in melanoma therapy and food processing [17,18]. Both carbazole and thiazole derivatives have a variety of biological properties, such as antimicrobial [19,20], anticancer [21,22], antidiabetic [23], anticonvulsant [24], and antioxidant properties [25,26].

Taking into account the above properties of the carbazole derivative and thiazole, we decided to design new dyes with an extensive structure. The required electron-acceptor properties were obtained via the phenyl group substituted in the 4-position by F, Cl, and Br atoms. Taking into account the values of Hammet’s constants, these halogen atoms are electron-withdrawing in nature and attract electron density towards themselves and away from other adjacent atoms. Furthermore, substitution of F, Cl, or Br for H on an aromatic ring provides the opportunity to study the heavy atom effect. As far as we know, carbazole–thiazole hybrids are very poorly understood. Therefore, the aim of our research is to expand knowledge about their spectroscopic properties and the scope of application of these materials in optoelectronics and in the design of anticancer drugs based on the tyrosinase inhibition mechanism. To achieve this goal, we continue existing research on carbazole–thiazole hybrids and investigate their photophysical and biological properties as well as their tyrosinase inhibition ability. Finally, the mechanism of inhibition and molecular docking to the active site of Concanavalin A (ConA) and bioavailability effects for all compounds were evaluated. In addition, the C-Cl molecule was presented in previous studies [27], discussing the influence of chlorine substituents on optical and biological properties of potential fluorescent probes. For the purposes of comparison of the influence of heavy atoms on the discussed properties, some of these results were intentionally repeated. The absolute novelty regarding this compound in this work include results concerning the following: development of a new, one-pot tricomponent method of its synthesis and testing of its ability to inhibit tyrosinase as well as the type of inhibition mechanism. It should also be noted that the presented studies are an introduction to further in vitro and in vivo studies. The newly synthesized carbazole derivatives in this study are tested for their suitability as fluorescent markers and verified against formal requirements. In turn, studies involving cell lines, where they will already constitute proper fluorescent markers, will be presented in the next stage of our work.

## 2. Materials and Methods

All details regarding chemical synthesis, UV-Vis absorption measurements and analysis [28], the mushroom tyrosinase inhibition assay [29,30], kinetic analysis of the inhibition of tyrosinase [23,31,32], and the computational procedure [33,34,35,36,37,38,39,40,41,42,43,44,45,46,47,48,49,50,51,52,53,54,55,56,57,58,59,60] and abbreviations have been placed in the Appendix A.

## 3. Results and Discussions

### 3.1. Chemical Synthesis

The titled carbazole-based thiazoles (C-Cl, C-Br, and C-F) were prepared in a simple and facile one-pot three-component reaction of appropriate benzaldehyde (**C**), thiosemicarbazide, and the corresponding 4-substituted 2-bromoacetophenones in anhydrous ethanol solution under reflux for 20 h. This simple reaction proceeds with greater than 99% yield without the use of an acid catalyst [61]. The synthesis of target compounds C-Cl, C-Br, and C-F is outlined in Figure 1. All obtained products were purified on silica gel column chromatography and fully characterized employing spectroscopic methods, ^1^H, ^13^C NMR, and ESI-HRMS analysis (see Appendix A). Characteristic signals from the thiazole ring-5H protons (at about 8.18 ppm), hydrazine NH groups (at about 12.32 ppm), the C-NH group (at about 170 ppm), and peaks corresponding to their molecular [M+H]^+^ ions indicating the formation of desired products were observed in the NMR and ESI-HRMS spectra.

### 3.2. Optoelectronic Characteristics

The steady-state fluorescence and absorption spectra of the tested dyes were firstly recorded in chloroform (Figure 1). The compounds show absorption spectra characterized by a relatively strong band with maximum at ca. 363–366 nm in this solvent, which are similar in shape and intensity. These long wavelength bands are almost structureless and have relatively high extinction coefficients (24,100–27,500 M^−1^cm^−1^), which is characteristic for π → π* transitions (Table 1). The other bands are visible in the range from 280 nm to 300 nm and are connected with n → π* transitions. The very similar absorption spectra observed for all tested compounds indicate that the nature of the substituents at position 4 of the phenyl ring does not interfere with the S_0_ → S_1_ transition.

Likewise, the fluorescence lifetimes and emission spectra of the dyes were evaluated in chloroform. The results are given in Figure 1 and Table 1. All dyes exhibit fluorescence with a maximum at 454 nm. Studying the influence of substituents on the fluorescence spectra, it is clear that its maximum does not shift upon replacing the substituent with a weaker EWG (electron-withdrawing group) part (C-Br → C-Cl → C-F); however, this effect is accompanied by an increase in the fluorescence intensity. The fluorescence quantum yield values are diminished from 11.37% to 10.98% and 7.33% for C-F, C-Cl, and C-Br derivatives, respectively. This phenomenon could result from internal conversion or intersystem crossing induced by molecular motion and the heavy atom effect. We hypothesize that the substitution of fluorine (F) with bromine (Br) enhances the probability of all S → T transitions, implying that intersystem crossing is more rapid than fluorescence for this molecule. The fundamental photophysical data are summarized in Table 1.

Figure 2 shows the fluorescence decay curve for C-Br in chloroform deconvoluted using FAST software (ver. 3.5.0). The fit to double-exponential functions was based on discrete component analysis. The same methodology is applied for fitting the other dyes in tested solvents.

The fluorescence lifetimes measured via a time-correlated single-photon counting method are shown in Table 1 above. In the C-F compound, τ_Fl_ is 1.234 ns in CHCl_3_. This value goes up to 1.261 and 1.265 ns, respectively, when the C-Cl and C-Br group, respectively, is present. Hence, the rate constant of radiative *k_r_* deactivation decreases from 9.21 × 10^7^ to 8.71 × 10^7^ and 5.79 × 10^7^ s^−1^, respectively, due to the stronger EWG nature of the substituent. Additionally, the data presented in Table 1 indicate that for the tested compounds, the nonradiative transition rates are two orders of magnitude greater than the radiative ones, suggesting the involvement of the excited singlet state that deactivates via internal conversion processes.

Figure 3 and Appendix A present the electronic absorption and fluorescence spectra of the tested compounds in different solvents. The influence of solvent polarity on the absorption is ambiguous, e.g., the wavelength of the long-wave band in THF is 360 nm and in DMSO it is 364 nm, i.e., the bathochromic effect occurs; but comparing its position in toluene and DMSO (365 nm and 364 nm), the opposite effect occurs for C-F (Table 1). In terms of fluorescence, a regular bathochromic shift of the band maximum is observed with increasing solvent polarity. This spectral effect is more pronounced, and, for example, changing the solvent from toluene to DMSO shifts the C-F fluorescence maximum from 428 nm to 489 nm, which indicates a greater stabilization of the excited singlet state in polar solvents. This phenomenon is characteristic of derivatives with increased dipole moments and charge-transfer (CT) character in the excited state [59,61].

Analyzing the correlation between the Stokes shift and the solvent polarity parameter, *f* (ε,n), a linear relationship is observed as shown in Appendix A. However, no reasonable correlation is found between the absorption peak position and the solvent polarity. This is not surprising, since the absorption maximum is mainly sensitive to the solvent polarizability related to the refractive index, and not to the permanent dipole moment of the solvent molecules contained in the dielectric constant; dielectric relaxation occurs during the lifetime of the excited state. This behavior is evident from the relationship between the absorption band position and the refractive index (*f* (n)), which is linear (Appendix A).

### 3.3. Tyrosinase Inhibition

The halogenated carbazoles C-Cl, C-Br, and C-F were examined for their inhibitory activity on the tyrosinase enzyme using the dopachrome method (Table 2) and were compared with standard inhibitors such as kojic acid (KA) and ascorbic acid (AA).

From the experimental data, it appears that all of the synthesized compounds have a higher effect of tyrosinase inhibition than AA (IC_50_ 386 µM). The results revealed that compound C-F is the most potent with a IC_50_ value 68 µM, and it exhibits a 1.06-times higher inhibitory effect than KA (IC_50_ 72 µM) and is almost 6-times more potent than AA (IC_50_ 386 µM). The fluorine atom present in this molecule is responsible for the high ability to inhibit tyrosinase. A slightly lower tyrosinase inhibition effect, compared to KA, was shown for compound C-Cl (IC_50_ 86 µM) containing a chlorine substituent. The compound C-Br with the bromine atom was characterized by the lowest activity among the tested compounds; however, it was 3.66-more potent than the activity of AA.

According to Figure 4, the Lineweaver–Burk plot showed that all carbazoles are characterized by a mixed type of inhibition which means that the inhibitor can bind to the enzyme regardless of whether the enzyme has already bound the substrate or not. An example for the most active compound C-F is presented in Figure 4.

The lowest value of the inhibition constant, K_m_ = 0.2896, which indicates the strongest affinity for the tyrosinase, was exhibited for the compound C-Br containing a bromine substituent in the 4-position in the phenyl ring.

### 3.4. Theoretical Sections

#### 3.4.1. Chemical and Physical Characteristics

The charge-transfer (CT) excitation for the tested compounds primarily corresponds to the HOMO → LUMO transition (Figure 5). The HOMO electrons are delocalized across the entire molecular framework, whereas the LUMO electrons are predominantly localized on the thiazole ring, the π-electron bridge, and the benzene ring attached to the carbazole unit. The substitution of the halogen group does not induce any significant shifts in the positions of the frontier orbitals. Additionally, the energy gap (${∆E}_{GAP}$, Appendix A) between the HOMO orbital and LUMO of the tested compounds shows only minor variations. In a GP (gas phase), the smallest ${∆E}_{GAP}$ value is characteristic of compound C-Br, while the largest is observed for compound C-F, with a difference of 0.0236 eV. As the solvent polarity increases, there is a consistent increase in ${∆E}_{GAP}$, with the C-F derivative exhibiting the highest value in water, while C-Cl has the lowest. A higher electronic chemical potential suggests an increased propensity for electron donation from the equilibrium system. The negative values of the chemical potential across all cases indicate spontaneous interactions with other compounds. The electrophilicity index indicates that C-Br is the most electrophilic, whereas C-F is the least electrophilic among the studied molecules. The tested derivatives are characterized by low chemical hardness, classifying them as soft molecules with high reactivity. The elevated electronegativity values imply an ease in forming covalent bonds during various chemical reactions.

To predict the reactive zones for electrophilic (red and red–yellow sites, negative) and nucleophilic (blue sites, positive) attacks of the investigated conformers, the Molecular Electrostatic Potential (MEP) surfaces were evaluated (Figure 5). For all dyes, the most electropositive region is located at the nitrogen atom of the *π*-electron bridge with the attached hydrogen atom, whereas the most electronegative region is found on the pyrrole moiety of the carbazole structure. For C-F, an additional electronegative region is the F atom. Based on this electrostatic distribution, it can be inferred that these derivatives will not have a significant tendency to act as hydrogen bond donors.

The spectral properties discussed below include one strong absorption and emission band, corresponding to the π-π* transitions (HOMO → LUMO photoexcitation). However, non-negligible contributions from other orbitals may also occur. To elucidate the nature of these electronic states, the density variation upon photoexcitation (Δ*ρ* (r)) computed for the first electronic transitions is graphically depicted in Figure 5**.** The Δ*ρ* (r) plots reveal that the density depletion zones (blue) are primarily delocalized on the thiazole ring, the *π*-electron linker, and the carbazole moiety. Conversely, the sites of density increment (purple) are scarcely visible on the carbazole part and the outer benzene ring for all molecules. Furthermore, the solvent polarity markedly impacts the parameters characterizing Δ*ρ* (r) (Appendix A). For each derivative, *D*_CT_ is reduced with increasing solvent polarity and the difference ${∆D}_{CT}^{vaccum-water}$ is 1.825 Å, 1.793 Å, and 1.1985 Å for C-Cl, C-Br, and C-F, respectively. For all derivatives, the *D*_CT_ indicates the CT character and confirms the contributions from the HOMO → LUMO transition; although, minor contributions from other orbitals should be expected. In particular, they can come from the transitions HOMO: HOMO → LUMO+1 with a participation of 3% for C-Cl and C-Br and 2% for C-F. The quantity of transferred charge diminishes monotonically as a function of solvent polarity. The greatest difference, Δ*q*_CT_, between the extreme environments (vacuum and aqueous phase) is observed for C-Br, amounting to 0.194 e, while the smallest is for C-F (0.181 e). Furthermore, the C-Br molecule exhibits the highest value of *q*_CT_. Similar to the transferred charge amount, the C-Br molecule also shows the highest charge-transfer distance, whereas C-F has the lowest. For each derivative, *D*_CT_ decreases with increasing solvent polarity, with the difference ${∆D}_{CT}^{vaccum-water}$ being 1.825 Å for C-Cl, 1.793 Å for C-Br, and 1.1985 Å for C-F.

Based on the free energy of solvation (Δ*G_solv_*, Appendix A), all molecules are well soluble in the tested solvents, and the Δ*G_solv_* is lower than −20 kcal/mol. Notably, the transition from DMSO to the most polar environment is associated with a marked increase in this parameter, indicative of the limited solubility of the tested dyes in aqueous media.

#### 3.4.2. Spectral and NLO Characteristics

The calculated absorption maxima ($\lambda_{max}^{Abs}$) established for tested molecules are presented in Table 3 and Appendix A. Only the PBE0 functional was considered in this study, as it gives the best concordance of the linear properties of many classes of compounds with respect to the values measured experimentally [38,39,40,41,42,43]. In this case, for C-Cl the average error is 3.13 nm and 3.44 nm for the vertical and cLR values, respectively. For C-Br and C-F, the error values are 4.99 nm and 5.36 nm and 3.75 nm and 3.32 nm, respectively. Moreover, the values obtained with the participation of PBE0 are slightly higher than those measured, with the exception of DMSO, where the inverse relationship is observed. Comparing the applied calculation models for the evaluation of the excitation energy, the values obtained in the cLR approximation are slightly higher in weakly polar solvents and lower in polar media compared to the vertical values. For this relation, the mean error is 0.93 nm. Moreover, as for the experimental values, the position of the $\lambda_{max}^{Abs}$ depends on the polarity of the medium. When evaluating the values obtained using the PBE0 functional, the tested dyes exhibit a non-monotonous behavior as a function of medium polarity for both vertical excitation energies and experimentally measured values. For compound C-Br, the theoretical data suggest a nearly monotonous increase in excitation energy with the increasing polarity of the surrounding environment. On the other hand, for the cLR value, a monotonous increase in the excitation energy is observed. The position of $\lambda_{max}^{Abs}$ is significantly influenced by the presence of halogen atoms. In terms of increasing $\lambda_{max}^{Abs}$ values, the tested molecules in the GP can be ordered as C-Br → C-Cl → C-F. However, in polar solvents, this sequence is disrupted to C-Cl → C-F → C-Br. These analyses, coupled with MEP simulations, substantiate the potential for specific interactions between the tested and solvent molecules. This is attributed to greater polarization and enhanced stabilization of the ground state (S_GS_) in polar solutions, which results in elevated excitation energy. Nevertheless, this observation is inconsistent with the polarity of the excited state (${\Delta \mu }_{CT-GS}$, Table 3 and Appendix A). Firstly, all examined dyes exhibit a μ_CT_ > μ_GS_ relationship across all solvents, indicative of positive solvatochromism. Additionally, *μ*_GS_ values increase monotonically as a function of solvent polarity. In contrast, *μ*_CT_ for C-Cl displays non-monotonous behavior, with a difference ${\Delta \mu }_{CT-GS}^{GP-H_{2}O}$ of 0.71 D. In the case of C-Br and C-F, a monotonous increase in *μ*_CT_ is observed. The most polar CT state in the gas phase is characteristic for C-Cl (6.41 D). During the transition from vacuum to solvent, the presence of bromine maximizes the polarity of the CT state, and in water, water ${\Delta \mu }_{CT-GS}^{GP(C2-C1)}$ and ${\Delta \mu }_{CT-GS}^{GP(C2-C3)}$ are both 3.06 D. Furthermore, for C-Cl, a monotonous decrease in ${\Delta \mu }_{CT-GS}$ is observed as a function of environment polarity, whereas for C-Br and C-F, a non-monotonous relationship is evident. Generally, the lowest-lying CT state of the evaluated derivatives is characterized by relatively weak polarity. These findings suggest that pure electrostatic interactions should not dominate solute–solvent interactions, and short-range specific interactions, such as hydrogen bonding and self-aggregation, may be present.

Table 3 and Appendix A present the values of the de-excitation energy ($\lambda_{max}^{Fl}$). Similar to the experimental values, the $\lambda_{max}^{Fl}$ values obtained with both PBE0 and cLR increase monotonously as a function of the medium polarity. This confirms that the tested derivatives have a charge-separated ground state and neutral excited state as a result of the solute–solvent interactions. As a result, a molecule in a neutral excited state remains electrically neutral, even though it is in a higher energy state. Energy absorption causes an electron to be transferred from a lower to a higher orbital (e.g., from HOMO to LUMO), but the overall charge in the molecule remains unchanged. This state is crucial in processes such as fluorescence, allowing the emission of energy in the form of light without altering the molecule’s total charge, while also affecting its chemical and physical properties, such as reactivity and spectroscopic behavior. Taking into account the change in the halogen atom, the tested dyes can be arranged in the following way in terms of shifting the position of the fluorescence maximum: C-F → C-Br → C-Cl. A greater bathochromic shift in the presence of a fluorine atom relative to the C-Cl molecule occurs in GP (9.23 nm) and in MeOH (7.75 nm) for vertical values. In the case of the values obtained within the cLR approximation, the greatest shift is observed in MeOH (12.59 nm). The exchange of chlorine to bromine does not change these solvent relationships, and the discussed differences are 10.84 nm; 13.85 nm; and 13.94 nm, respectively. The theoretically determined $\lambda_{max}^{Fl}$ values are bathochromically shifted compared to the experimental ones, except for the C-Cl maximum in toluene, where the hypsochromic effect is observed. Comparing the fluorescence maxima calculated with the measured values, the largest mean error was obtained for the C-Cl derivative, which is 27.67 nm and 27.55 nm for the vertical and cLR values, respectively. The magnitude of this error decreases significantly in the presence of bromine and fluorine and amounts to 5.078 nm and 11.76 nm for C-Br and 2.61 nm and 10.25 nm for C-F.

Figure 6 presents the theoretical absorption and fluorescence bands of the studied molecules in chloroform in graphical form. Both the theoretical and experimental absorption (Figure 1) spectra indicate the presence of additional low-intensity peaks within the same range. These peaks result from electronic transitions to higher excited states, which have a lower probability of occurrence (hence their lower intensity). These are so-called spin-forbidden transitions or transitions between less favorable energy states, as confirmed by the agreement between theory and experiment. In the case of the C-Br spectrum, due to the presence of bromine (heavier than chlorine), these peaks are shifted more significantly toward the red (C-Br: 326.16 nm, 317.41 nm, and 313.87 nm; C-Cl: 325.73 nm, 314.99 nm, and 313.43 nm) and show greater differences in intensities. Similarly, as with C-Cl and C-Br, the less intense peaks in the theoretical spectrum of C-F (326.27 nm, 313.61 nm, and 303.58 nm) correspond to real transitions between higher excited states. Fluorine-containing compounds as substituents may exhibit better theoretical–experimental agreement at lower energies due to smaller spin–orbit coupling differences compared to bromine or chlorine. Differences in the intensities of the additional peaks, hypsochromically shifted relative to the main band (*π*-*π** transition), between experiment and theory arise because theoretical calculations often assume ideal conditions and omit certain experimental factors, such as interactions of the molecules with the solvent or other molecules in the sample. Furthermore, both the theoretical and experimental spectra support the earlier hypothesis regarding the influence of other orbitals on the absorption bands of the molecules under study. The theoretical (Figure 6) and experimental (Figure 1) fluorescence spectra are in very good agreement, with no additional low-intensity peaks, suggesting that fluorescence occurs from a single dominant excited state to the ground state. For C-Br, the red shift in fluorescence is due to the greater energy difference between the excited state and the ground state for the bromine-containing compound. Excellent agreement between experiment and theory is also observed for C-F, where fluorescence is clearly associated with a single excited state, with no additional fluorescence transitions. The analysis of the spectrum in chloroform yields analogous conclusions when other solvents are used.

The presented dyes are described by relatively high Stokes shift values ${(\Delta \nu }^{St}$). These values for all compounds increase monotonously in the environment polarity function. When changing the environment from gas to water (GP → H_2_O transition), the changes are from 3105.90 cm^−1^ to 7057.13 cm^−1^, from 32,965.39 cm^−1^ to 6859.64 cm^−1^, and from 3697.63 cm^−1^ to 7235.20 cm^−1^ for C-Cl, C-Br, and C-F, respectively. Against this background, combining the experimental analysis of spectroscopic properties with theoretical data, it can be conclusively stated that the newly synthesized carbazole molecules fulfill all the criteria for fluorescent probes.

For many years, experimental and theoretical studies have been conducted all over research centers to analyze the nonlinear optical properties (NLO) of compounds exposed to intense laser light. The analyzed properties, such as the polarizability ($\langle \alpha \rangle$) and the first hyperpolarizability (*β*_vec_), allow for determining the interrelationship of nonlinear properties with the electronic structure to design new multifunctional compounds, e.g., fluorescent probes. The theoretically determined values of NLO are presented in Table 3 and Appendix A. The obtained data indicate a monotonous increase in the value of $\langle \alpha \rangle$ with an increase in the polarity of the medium. The derivative substituted with a bromine atom has the highest value, while the one with a fluorine atom has the lowest, with ${∆\langle \alpha \rangle }^{C2-C3}$ being 24.86 a.u. in GP and rising to 35.10 a.u. in water. In the case of the *β*_vec_, the values obtained for the chloro-substituted derivative decrease as a function of the solvent polarity, while for C-Br and C-F, they increase. The C-F derivative is characterized by the highest *β*_vec_ values in polar centers and C-Cl in weakly polar solvents. At the same time, these relationships indicate that from the point of view of the search for new multifunctional markers, the presence of the fluorine atom will be a factor that maximizes the NLO response of the system.

Table 3 and Appendix A show the values of two-photon absorption cross section (TPA) values based on the CAM-B3LYP functional. For all compounds, both the values expressed in GM ($\sigma_{OF}^{\left(2\right)}$) and in a.u $\langle \delta^{OF}\rangle$ decrease monotonously with increasing medium polarity. Moreover, $\langle \delta^{OF}\rangle$ and $\sigma_{OF}^{\left(2\right)}$ are relatively low and highlight the limitations of the evaluated molecules in studies utilizing two-photon absorption. The decrease in TPA value is observed up to DMSO, and the transition to more polar solvents does not contribute to a further minimization of the described properties. However, the bromine substituent is a factor in maximizing the TPA value.

#### 3.4.3. Biological Characteristics

Nowadays, the search for more and more efficient fluorescent markers, used in various fields of science, especially in medical imaging, is underway. Such markers, in order to be used in bioimaging, must have a characteristic functional group that allows an easy and quick conjugation with the protein [61]. As shown earlier, the tested compounds are not suitable for TPA imaging; however, introducing into their skeleton a suitable functional group, such as -CHO, -COOH (conjugation with the -NH_2_ groups of the protein, ConA in our study), maleimide, or pyridyl disulfides (conjugation with the protein through the -SH groups, HSA) can be a great tool in single-photon imaging. The performed AutoDock simulations showed that all derivatives have the highest affinity for LYS116 (Figure 7) during conjugation with ConA. It is, therefore, the site of the greatest probability of conjugation through the amino group of the macromolecule. In all instances, the binding energy (Δ*G_b_*) is −5.5 kcal/mol (Appendix A), and the active binding site exhibits inhibition constants (K*_i_*) of 0.78 μM for C-Cl, 0.89 μM for C-Br, and 0.82 μM for C-F. The carbazole molecules are positioned within an aromatic cage formed by VAL187, LYS116, VAL188, THR120, GLU122, and where no *π-π** interactions occur, and the system lacks stabilization by hydrogen bonding. During the interaction with HSA, the binding center for C-Cl is CYS448, while for C-Br and C-F, it is CYS438. As the dyes adjust spatially, they become surrounded by additional amino acids, with C-F demonstrating the highest affinity (Appendix A), evidenced by a Δ*G_b_* of −9.8 kcal/mol and a K*_i_* of 0.85 μM. For C-Cl, Δ*G_b_* is −9.4 kcal/mol and K*_i_* = 0.75 μM, and for C-Br, Δ*G_b_* is −9.5 kcal/mol and K*_i_* = 0.93 μM. The biocomplexes formed at this active site are not stabilized by additional interactions in the dye–amino acid system. It is worth mentioning that in this case, both CYS438 and CYS448 are active binding centers with the highest probability of interaction in the protein–marker system. For C-Cl and C-Br, ${∆∆G}_{b}^{LYS-CYS}$ is only 0.8 kcal/mol, while for C-F it is 1.0 kcal/mol. In the process of adjusting to the aromatic cavity, in each case, all rotations occur on the bonds of the π-electron bridge, by changing the angles C=N-N and N-N-C. The largest geometric changes during the fitting were observed for the C-F-LYS derivative, where the change in the dihedral angle of C=N-N was 0.04851° and −0.04438° for N-N-C. In turn, for the C-F-CYS system, the changes in the values of these angles amount to 0.03053° and −0.01308°, respectively. For each investigated derivative, significantly greater changes in the dihedral angles of the π-electron bridge occur during the adjustment to the ConA active center, and the mean ${\Delta ∢}^{LYS-CYS}$ differences are 0.02198° for C=N-N and 0.01494° for N-N-C. Changes in the described angles indicate slight structural changes in the presented molecules during the formation of an active biocomplex with the protein.

The evaluated carbazole dyes exhibit relatively good bioavailability. Notably, C-Cl shows the highest LogP value at 8.56, whereas C-F displays the lowest at 7.89. This suggests that the analyzed derivatives possess excellent permeability through cell membranes, a highly desirable attribute in drug design. The theoretical LogBCF values, ranging from −5.23 for C-F to −5.84 for C-Br, indicate a lack of bioaccumulation in the tissues of living organisms, facilitating ease of excretion via urine. Consequently, these molecules should not bioaccumulate post their optical role fulfillment. Moreover, all compounds demonstrate high metabolism by CYP450-2D6 (probability > 73%) and CYP450-3A4 (probability > 79%) (Appendix A). This suggests that, irrespective of the substituent attached, all derivatives will be rapidly excreted from tissues without interacting with other drugs and biomolecules.

The evaluated markers should be regarded as compounds that are safe for human use. The calculated LD50 values for the intraperitoneal route of administration are 1302.00 mg/kg for C-Cl, 1073.00 mg/kg for C-Br, and 765.20 mg/kg for C-F. For the intravenous route, the LD50 values are 172.00 mg/kg, 166.80 mg/kg, and 193.60 mg/kg, respectively. For oral administration, the values are 4387.00 mg/kg for C-Cl, 6112.00 mg/kg for C-Br, and 1587.00 mg/kg for C-F. For the subcutaneous route, the LD50 values are 1353.00 mg/kg, 3019.00 mg/kg, and 2203.00 mg/kg, while for the intraperitoneal route, they are 1973.50 mg/kg, 1875.00 mg/kg, and 1381.00 mg/kg, respectively. Furthermore, all tested derivatives are non-carcinogenic, with the probability of occurrence ranging from 55% to 59%, non-immunotoxic (75–78%), and non-cytotoxic (70–74%). However, the tested dyes show mutagenicity with the highest probability of occurrence for C-F (64%) and the lowest for C-Cl (58%). There is also a probability of hepatotoxicity of 62% for C-F and 57% for C-Cl.

Beyond their primary attributes, the evaluated derivatives exhibit a variety of biological activities, indicating their potential utility in multiple medical domains (Appendix A). Notably, all molecules, regardless of the substituent’s nature that is attached to their structure, display significant analgesic properties. They are effective against Awasky disease, herpes simplex virus, Rift Valley fever, and adenovirus. Additionally, these compounds exhibit antiarrhythmic properties and demonstrate activity against both the Hong Kong influenza virus and influenza A. Their antipsychotic effects are notable at the diazepine site, and they show antitumor activity through the inhibition of cyclin-dependent kinase 4. Furthermore, these derivatives act as gamma-radioprotectors via mechanism II, inhibit HIV1-protease and human factor XA, and possess tuberculostatic properties by inhibiting dihydrofolate reductase. In the mentioned cases, the derivative C-F is characterized by the lowest value for the occurrence of a given biological activity. Moreover, the presence of a fluorine atom in the structure of carbazole derivatives is a factor limiting the occurrence of biological and medical properties. For example, the probability of anti-influenza A activity for C-Cl is 96%, for C-Br it is 99%, and for C-F it drops to 0.02%. Similar dependencies can be found in antitumor DNA antimetabolitic activity, antitumor topoisomerase II inhibitory activity, LOX inhibitory activity, and others. The type of attached halogen atom also has an influence on the occurrence of a given biological property. Only C-Br has antibacterial properties at the level of 66%. The presence of a bromine atom also contributes to the occurrence of Anti-Influenza Birds activity (26%), COX1 inhibitory activity (98%), and Progestagenic activity (93%). In the presence of a fluorine atom, the derivative has characteristic properties such as Acyl-CoA-holesterol transferase inhibitory activity (66%), anti Issyk-Kul hemorrhagic fever activity (17%), and antiencephalitic activity (93%). For C-Cl with a chlorine atom, vasorelaxant activity (49%) is the most characteristic property.

## 4. Conclusions

We have introduced three newly synthesized biologically active chromophores, carbazole–thiazole-based dyes, with their comprehensive spectroscopic characterization, starting from the quantum-chemical calculations approach and theoretical description of the considered molecules up to the tyrosinase inhibition evaluation. The findings illustrate that these materials function as highly effective instruments for the design and creation of fully organic fluorescent markers. Moreover, the spectroscopic characteristics of these dyes can be readily adjusted and anticipated by making slight modifications to their chemical structure. Spectroscopic studies have shown that replacing the substituent with a weaker electron-withdrawing group (C-Br → C-Cl → C-F) does not affect the position of the absorption and fluorescence bands, but only causes a slight increase in fluorescence intensity. The fluorescence quantum yields are 11.37%, 10.98%, and 7.33% for C-F, C-Cl, and C-Br derivatives in chloroform, respectively. Additionally, an increase in solvent polarity results in a significant red shift of the fluorescence maximum and an increase in the Stokes shift which indicates the excited state might be considered as an intramolecular charge transfer state. The analysis of NLO parameters showed that from the point of view of the search for new multifunctional markers, the presence of the fluorine atom will be a factor that maximizes the nonlinear response of the system, and the presence of a bromine atom is a factor in maximizing the TPA value. Our UV–Vis spectroscopic results indicate also that carbazole–thiazoles exhibit high tyrosinase inhibitory activity, comparable to the activity of kojic acid and are several-times more active than ascorbic acid, which suggests that they can be used in melanoma therapy and food processing. The presented studies clearly indicate the application and usefulness of the tested derivatives as fluorescent markers in in vitro and in vivo studies. The tested probes show high affinity to proteins, easily forming biocomplexes, showing no bioaccumulation and no interactions with other biomolecules and drugs. In addition, the high LD50 value at various administration routes makes them safe for living organisms and allows them to be used as valuable pharmaceutical preparations in various fields of medicine. The comprehensive experimental and theoretical investigations unequivocally confirm that the newly synthesized carbazole derivatives are exceptional candidates for advanced pharmacological, cancer therapy, and fluorescent markers.

## Data Availability

The data presented in this study are available on request from the corresponding author.

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
