# Peer review of "Synthesis of Carbazole–Thiazole Dyes via One-Pot Tricomponent Reaction: Exploring Photophysical Properties, Tyrosinase Inhibition, and Molecular Docking"

_sensors, 2024, doi:10.3390/s24196368_

Round 1

Reviewer 1 Report

Comments and Suggestions for Authors

Krawczyk et al. present a study of three new carbazole-based thiazoles and the optical properties, electronic structures and biological inhibition properties are evaluated. I think the study is of interest to the readers of “Sensors”, but on several points I would encourage the authors to clarify and reconsider their arguments, hence I recommend publication after minor revision. In particular, I think there is a missing link between the different parts of the work i.e. photophysical analysis, theoretical calculations and biological evaluation. It would be great if e.g. the theoretical insights could be more directly incorporated in the discussion in the part on photophysical and biological properties. Also, throughout the manuscript it is mentioned that the carbazole-based thiazoles are fluorescent markers for bioactivity. I completely agree that C1-C3 are fluorescent compounds, I, however, fail to see the link between fluorescence properties and the investigated biological evaluation. In the method-section in the supplementary information, it does not seem like the fluorescence properties are used to determine the biological activities. Please clarify.

Do you have any insights about a carbazole-based thiazole, where R = H? The motivation for using halogens is the heavy atom effect, but then it would be interesting to evaluate if it indeed has any influence. Also, the investigated biological activities evaluated and described on page 12 do not seem to have anything to do with emission properties? Rather a size argument could be relevant, in particular the drop in activity for C3 (e.g. line 396) could be due to the smaller size of F compared to Cl and Br? Here a comparison to R = H would also be interesting.

How come the naming of the carbazole-based thiazoles is not following the periodic table i.e. C1 (F), C2 (Cl) and C3 (Br) or C1 (Br), C2 (Cl) and C3 (F)? Then the discussed trends, e.g. on page 3, line 106-112, would be much easier to follow for the reader. I constantly had to go back to scheme 1 to remind myself about the naming which seems unpractical for the greater understanding of the study, especially when referring to trends. One could even consider a naming of C-F, C-Cl and C-Br increasing the understanding of the text significantly.

Absorption spectra in general: Information about the extinction coefficients is lost by normalizing the absorbance spectra; hence I would encourage the authors to show the y-axis as molar absorptivity in all absorption spectra. By doing so, it is also immediately visible if the change of substituents affects the allowance of the π-π* transitions.

What is the unit used for the fluorescence spectra in Figure 1? Is it corrected with Einstein coefficients (maybe λ5 instead of v5?) to display the stimulated emission instead of the spontaneous emission? If it is the case, please explain why and add this explanation either as a footnote somewhere or in the description in the supplementary information. I would, however, say it is common practice to simply show the measured (spontaneous) emission.

Table 1: I would encourage the authors to reconsider their use of significant decimals for all photophysical parameters. For absorption and emission maxima, I would not expect a certainty to more than one wavelength (and not decimal wavelengths as indicated by e.g. 489.4 nm for C1 in DMSO). For quantum yield and extinction coefficients: Is it fair to have to significant decimals? For lifetimes: Is three significant decimals reasonable? Why is the extinction coefficient in toluene not determined?

Page 3, line 90-94:  Please define λ(abs, max) before using it. The maximum absorption band is observed around 260 nm and not 365 nm in Figures 1 and 3, so there seems to be a selective definition for λ(abs, max).  What is “All of them are almost structureless” referring to? The band at 365 nm or all bands in the absorption spectrum?

Figure 2: It would be useful if figure 2 showed a comparison of the kinetic traces of C1-C3, so that it would be easier to evaluate the mono- vs. biexponential fitting. Also, what is the physics/chemistry behind using biexponential over monoexponential fitting functions or is it purely a mathematical consideration? Could e.g. different rotamers or close-lying emissive states explain biexponential decay behavior?

Figure 3: It is very interesting that a shoulder is observed in fluorescence spectra for all tested dyes in MeOH, and that for C1 and C3 it is a high energy shoulder, whereas for C2 it is a low energy shoulder – do you have any idea why? Can the calculations give any insights? Also, it would be useful if either the labels were sorted after polarity or if a polarity parameter (e.g. dielectric constant) would be mentioned somewhere. There seems to be opposite trends in polarity in the absorption and emission data, which could suggest that a purely polarity argument cannot be used as introduced on page 5, line 136-137. Maybe a Lippert-Mataga plot could give some insights? Hydrogen-bonding probably also plays a significant role for the optical properties of these compounds? Hydrogen-bonding is mentioned in the theoretical part of the work. This effect would be most significant in MeOH which could maybe be related to the emission shoulder observed in that solvent? The band width also seems to be very sensitive to the solvent, but evaluation on an energy scale (e.g. cm-1 or eV) would help that analysis. Why is the data obtained in chloroform not included in figure 3?

Would it be possible to do the photophysical analysis in DMSO/H2O mixtures to more closely resemble the conditions of those used in the tyrosinase inhibition experiments?

Page 6, line 157: “whether the enzyme has already bound the substrate.” It feels like a part of the argument is missing here. Whether the enzyme has bound the substrate or not? Or maybe the authors have some other mechanism in mind?

Figure 4: Please clarify what “dopa”, “100” and “50” in the caption refer to.

Please define “GP”.

Page 7-8, lines 191-193: To me it is contradicting to first claim that hydrogen bonding is not significant and then subsequently say that it has an effect. Please clarify.

Page 8, lines 207-208 and 218-219: This statement seems to be repeated.

Page 9-10, lines 226 – 263: This discussion of the calculated absorption properties would benefit from a comparison to trends seen for the experimental absorption properties.

Page 10, line 267: What does it exactly mean that the derivatives have a “neutral excited state”?

Page 10, line 284: What is a GPàH2O transition? Or does this simply mean the difference between the data obtained in the gas phase and in water?

Page 10, line 286-287: “From the aforementioned analysis, it can be conclusively stated that the newly synthesized carbazole molecules fulfill all the criteria for fluorescent probes.” Is this comment related to calculated parameters or? I mean, simply from the experimental data it should already be possible to conclude that the new compounds have fluorescence properties.

Page 11: How does these calculated parameters of e.g. binding energies compare to the experimental observations?

Page 11-12, lines 342-347: Are the differences in dihedral angles significant?

The sections in lines 348-358 and lines 359-369 on page 12 are repetitive stating the same information. Please modify.

Page 15, line 408: I would not say that the new carbazole-based thiazoles are biological themselves, rather they have a biological activity?

Some minor aspects:

Page 1, line 14: Replace “…ring, are a…“ with “…ring, and is a…”           

Page 1, line 18: Replace “… that carbazole…” with “…that carbazoles…”

Page 1, line 35: Replace “Carbazole ring is electron rich system…” with "The carbazole ring is an electron-rich system…”

Page 1, line 43: It would be useful to introduce a section break, when the focus changes from carbazole to tyrosinases i.e. the new section should begin with “Tyrosinase is a Cu-containing…”

Page 2, line 70: UV-Vis is just a part of the electromagnetic spectrum, but not exactly a technique, so please replace “, UV-Vis measurements and…” with “UV-Vis absorption measurements and..” .  Also, it looks like something has gone wrong with the font size in the “Materials and methods” section.

Page 2, line 73: Replace “…thiazole (C1-C3)…” with “…thiazoles (C1-C3)…”

Page 5, line 132: Replace “…presents…” with “…present…”

Table 2. Heading “Inhibitory Mechanizm” should be replaced with “Inhibitory Mechanism”.      

Table SI5: Some commas are wrongly shown as “,” rather than “.”

Author Response

  1. Krawczyk et al. present a study of three new carbazole-based thiazoles and the optical properties, electronic structures and biological inhibition properties are evaluated. I think the study is of interest to the readers of “Sensors”, but on several points I would encourage the authors to clarify and reconsider their arguments, hence I recommend publication after minor revision. In particular, I think there is a missing link between the different parts of the work i.e. photophysical analysis, theoretical calculations and biological evaluation. It would be great if e.g. the theoretical insights could be more directly incorporated in the discussion in the part on photophysical and biological properties. Also, throughout the manuscript it is mentioned that the carbazole-based thiazoles are fluorescent markers for bioactivity. I completely agree that C1-C3 are fluorescent compounds, I, however, fail to see the link between fluorescence properties and the investigated biological evaluation. In the method-section in the supplementary information, it does not seem like the fluorescence properties are used to determine the biological activities. Please clarify.

Answer:

In order to clarify the research presented and its objectives, the following fragment was added to the introduction:

It should also be noted that the presented studies are an introduction to further in vitro and in vivo studies. The newly synthesized carbazole derivatives in this study are tested for their suitability as fluorescent markers and verified against formal requirements. In turn, studies involving cell lines, where they will already constitute proper fluorescent markers, will be presented in the next stage of our work.

  1. Do you have any insights about a carbazole-based thiazole, where R = H? The motivation for using halogens is the heavy atom effect, but then it would be interesting to evaluate if it indeed has any influence. Also, the investigated biological activities evaluated and described on page 12 do not seem to have anything to do with emission properties? Rather a size argument could be relevant, in particular the drop in activity for C3 (e.g. line 396) could be due to the smaller size of F compared to Cl and Br? Here a comparison to R = H would also be interesting. Answer:

We agree with you, however, we failed to obtain a compound, where R = H. Our observations show that Hantzsch cyclization with unsubstituted bromoacetophenone proceeds with low yield and purity.

  1. How come the naming of the carbazole-based thiazoles is not following the periodic table i.e. C1 (F), C2 (Cl) and C3 (Br) or C1 (Br), C2 (Cl) and C3 (F)? Then the discussed trends, e.g. on page 3, line 106-112, would be much easier to follow for the reader. I constantly had to go back to scheme 1 to remind myself about the naming which seems unpractical for the greater understanding of the study, especially when referring to trends. One could even consider a naming of C-F, C-Cl and C-Br increasing the understanding of the text significantly .

Answer:

For better understanding of the text of the manuscript, in accordance with your suggestion, we changed the nomenclature to C-Cl, C-Br and C-F.

  1. Absorption spectra in general: Information about the extinction coefficients is lost by normalizing the absorbance spectra; hence I would encourage the authors to show the y-axis as molar absorptivity in all absorption spectra. By doing so, it is also immediately visible if the change of substituents affects the allowance of the π-π* transitions.

Answer: The authors agree with the Reviewer's opinion. However, our intention was to show the influence of the substituent as well as the solvent polarity on the position of absorption and fluorescence spectra. Taking into account the Reviewer's suggestion, the absorption spectra as a function of molar absorptivity vs. wavelength are added to the ESI file.

  1. What is the unit used for the fluorescence spectra in Figure 1? Is it corrected with Einstein coefficients (maybe λ5 instead of v5?) to display the stimulated emission instead of the spontaneous emission? If it is the case, please explain why and add this explanation either as a footnote somewhere or in the description in the supplementary information. I would, however, say it is common practice to simply show the measured (spontaneous) emission.

Answer:

In Figure 1, the absorption intensity is presented as A/n, while the emission intensity is represented as E/n^5 assuming that the measurement was done linear in wavelength. The scaled intensities versus frequency give a proper comparison for the different molecules. We have changed the X-asis from wavelength to frequency

  1. Table 1: I would encourage the authors to reconsider their use of significant decimals for all photophysical parameters. For absorption and emission maxima, I would not expect a certainty to more than one wavelength (and not decimal wavelengths as indicated by e.g. 489.4 nm for C1 in DMSO). For quantum yield and extinction coefficients: Is it fair to have to significant decimals? For lifetimes: Is three significant decimals reasonable? Why is the extinction coefficient in toluene not determined?

Answer: The number of significant decimals for the photophysical parameters listed in Table 1 has been corrected. In the case of extinction coefficients, the actual value is e.g. 23400 M-1 cm-1, so we do not provide any decimals.

The maximum extinction coefficient of the tested compounds in toluene was not determined due to too low solubility.

  1. Page 3, line 90-94: Please define λ(abs, max) before using it. The maximum absorption band is observed around 260 nm and not 365 nm in Figures 1 and 3, so there seems to be a selective definition for λ(abs, max). What is “All of them are almost structureless” referring to? The band at 365 nm or all bands in the absorption spectrum?

Answer: The authors agree with the Reviewer that the maximum absorption band is observed around 260 nm, not 365 nm as shown in Figures 1 and 3. The description refers to the long wavelength absorption band. The text has been corrected accordingly. λ(abs, max) is defined in Table 1.

  1. Figure 2: It would be useful if figure 2 showed a comparison of the kinetic traces of C1-C3, so that it would be easier to evaluate the mono- vs. biexponential fitting. Also, what is the physics/chemistry behind using biexponential over monoexponential fitting functions or is it purely a mathematical consideration? Could e.g. different rotamers or close-lying emissive states explain biexponential decay behavior?

Answer: The use of biexponential fitting functions over monoexponential ones was a purely mathematical consideration. Since the correlation coefficient of about 1.4-1.7 is somewhat too high, the number of exponents used to fit the fluorescence decay curves was checked. For several samples, the fluorescence decay curves were re-recorded over a longer time range.

The fluorescence decay curves were deconvoluted using FAST software. The fitting was based on discrete component analysis. A comparison of the kinetic traces of C-Br with  the mono- vs. biexponential fitting are shown below.

Figure Fluorescence decay profile for C2 measured in chloroform.; λex = 373 nm; λem = 460 nm.

As suggested by the Reviewer, different rotamers or closely spaced emission states may explain the biexponential decay behavior. According to this approach, the energies of all emitting states are rather very similar so that the fluorescence of the compounds is observed as one broadened band. It should be noted that all deactivation processes start from the planar conformation of the excited molecule and are associated with rotation about individual bonds.

  1. Figure 3: It is very interesting that a shoulder is observed in fluorescence spectra for all tested dyes in MeOH, and that for C1 and C3 it is a high energy shoulder, whereas for C2 it is a low energy shoulder – do you have any idea why? Can the calculations give any insights? Also, it would be useful if either the labels were sorted after polarity or if a polarity parameter (e.g. dielectric constant) would be mentioned somewhere. There seems to be opposite trends in polarity in the absorption and emission data, which could suggest that a purely polarity argument cannot be used as introduced on page 5, line 136-137. Maybe a Lippert-Mataga plot could give some insights? Hydrogen-bonding probably also plays a significant role for the optical properties of these compounds? Hydrogen-bonding is mentioned in the theoretical part of the work. This effect would be most significant in MeOH which could maybe be related to the emission shoulder observed in that solvent? The band width also seems to be very sensitive to the solvent, but evaluation on an energy scale (e.g. cm-1 or eV) would help that analysis. Why is the data obtained in chloroform not included in figure 3?

Answer: All three compounds tested in methanol as well as in DMSO are very weakly fluorescent (fluorescence quantum yield of about 0.1%). This causes that the recorded fluorescence was at the limit of the sensitivity of the device. The observed high (and low) energy shoulder is related to the device and not to the properties of the compounds tested. The fluorescence spectra were also recorded on Edinburgh Instruments FLS920P Spectrometer and the high (and low) energy shoulder is not observed. However, the spectra are not corrected, so there is some difference in the maximum wavelength.

Figure The fluorescence spectra recorded in MeOH on Edinburgh Instruments FLS920P Spectrometer

The correlation between spectral shifts and solvent polarity function is shown in Figure SI4 in the ESI file. A short description was added to the revised manuscript.

The authors agree with the Reviewer's comment that hydrogen bonding may also play a role in the optical properties of the compounds studied. However, more detailed analyses with a larger number of protic and aprotic solvents are needed to support such a conclusion and this was not the subject of this work.

The data obtained in chloroform are presented in Figure 1.

  1. Would it be possible to do the photophysical analysis in DMSO/H2O mixtures to more closely resemble the conditions of those used in the tyrosinase inhibition experiments?

Answer: We performed the measurements in DMSO/H2O mixtures, but due to the very poor solubility of the compounds in water and the negligible fluorescence, the obtained results are not very reliable and were not included in the manuscript.

Figure Changes in the position of absorption and fluorescence spectra after adding different amounts of water to DMSO solution.

  1. Page 6, line 157: “whether the enzyme has already bound the substrate.” It feels like a part of the argument is missing here. Whether the enzyme has bound the substrate or not? Or maybe the authors have some other mechanism in mind?

Answer: The misprint has been corrected.

  1. Figure 4: Please clarify what “dopa”, “100” and “50” in the caption refer to.

Answer: Done.

  1. Please define “GP”.

Answer: All abbreviations were presented in the SI. However, the appropriate note was added in section 2.

  1. Page 7-8, lines 191-193: To me it is contradicting to first claim that hydrogen bonding is not significant and then subsequently say that it has an effect. Please clarify.

Answer: The text has been edited.

  1. Page 8, lines 207-208 and 218-219: This statement seems to be repeated.

Answer: The text has been edited

  1. Page 9-10, lines 226 – 263: This discussion of the calculated absorption properties would benefit from a comparison to trends seen for the experimental absorption properties.

Answer: The comparison presented in the manuscript has been enriched with Figure 6 and the following text:

Figure 6 presents the theoretical absorption and fluorescence bands of the studied molecules in chloroform in graphical form. Both the theoretical and experimental absorption (Figure 1) spectra indicate the presence of additional low-intensity peaks within the same range. These peaks result from electronic transitions to higher excited states, which have a lower probability of occurrence (hence their lower intensity). These are so-called spin-forbidden transitions or transitions between less favorable energy states, as confirmed by the agreement between theory and experiment. In the case of the C-Br spectrum, due to the presence of bromine (heavier than chlorine), these peaks are shifted more significantly toward the red (C-Br: 326.16 nm, 317.41 nm, and 313.87 nm; C-Cl: 325.73 nm, 314.99 nm, and 313.43 nm) and show greater differences in intensities. Similarly, as with C-Cl and C-Br, the less intense peaks in the theoretical spectrum of C-F (326.27 nm, 313.61 nm, and 303.58 nm) correspond to real transitions between higher excited states. Fluorine-containing compounds as substituents may exhibit better theoretical-experimental agreement at lower energies due to smaller spin-orbit coupling differences compared to bromine or chlorine. Differences in the intensities of the additional peaks, hypsochromically shifted relative to the main band (π-π* transition), between experiment and theory arise because theoretical calculations often assume ideal conditions and omit certain experimental factors, such as interactions of the molecules with the solvent or other molecules in the sample. Furthermore, both the theoretical and experimental spectra support the earlier hypothesis regarding the influence of other orbitals on the absorption bands of the molecules under study. The theoretical (Figure 6) and experimental (Figure 1) fluorescence spectra are in very good agreement, with no additional low-intensity peaks, suggesting that fluorescence occurs from a single dominant excited state to the ground state. For C-Br, the red shift in fluorescence is due to the greater energy difference between the excited state and the ground state for the bromine-containing compound. Excellent agreement between experiment and theory is also observed for C-F, where fluorescence is clearly associated with a single excited state, with no additional fluorescence transitions. The analysis of the spectrum in chloroform yields analogous conclusions when other solvents are used.

  1. Page 10, line 267: What does it exactly mean that the derivatives have a “neutral excited state”?

Answer: The text has been enriched with the following fragment:

As a result, a molecule in a neutral excited state remains electrically neutral, even though it is in a higher energy state. Energy absorption causes an electron to be transferred from a lower to a higher orbital (e.g., from HOMO to LUMO), but the overall charge of the molecule remains unchanged. This state is crucial in processes such as fluorescence, allowing the emission of energy in the form of light without altering the molecule's total charge, while also affecting its chemical and physical properties, such as reactivity and spectroscopic behavior.

  1. Page 10, line 284: What is a GPàH2O transition? Or does this simply mean the difference between the data obtained in the gas phase and in water?

Answer: The text has been replaced as follows:

When changing the environment from gas to water (GP→H2O transition),

  1. Page 10, line 286-287: “From the aforementioned analysis, it can be conclusively stated that the newly synthesized carbazole molecules fulfill all the criteria for fluorescent probes.” Is this comment related to calculated parameters or? I mean, simply from the experimental data it should already be possible to conclude that the new compounds have fluorescence properties.

Answer: The text has been replaced as follows:

From the aforementioned analysis to Against this background, combining the experimental analysis of spectroscopic properties with theoretical data,

  1. Page 11: How does these calculated parameters of e.g. binding energies compare to the experimental observations?

Answer: Simulations using AutoDock are intended to indicate areas of active binding to a macromolecule. Unfortunately, we did not perform such experimental studies. At the same time, the binding energy results are not adequate and comparable with the results presented in section 3.3.

  1. Page 11-12, lines 342-347: Are the differences in dihedral angles significant?

Answer: The text has been replaced as follows:

Changes in the described angles indicate slight structural changes of the presented molecules during the formation of an active biocomplex with the protein.

  1. The sections in lines 348-358 and lines 359-369 on page 12 are repetitive stating the same information. Please modify.

Answer: The text has been edited.

  1. Page 15, line 408: I would not say that the new carbazole-based thiazoles are biological themselves, rather they have a biological activity?

Answer: The text has been replaced as follows:

From biological chromophores to biologically active chromophores 

Some minor aspects:

Page 1, line 14: Replace “…ring, are a…“ with “…ring, and is a…”        

Page 1, line 18: Replace “… that carbazole…” with “…that carbazoles…”

Page 1, line 35: Replace “Carbazole ring is electron rich system…” with "The carbazole ring is an electron-rich system…”

Page 1, line 43: It would be useful to introduce a section break, when the focus changes from carbazole to tyrosinases i.e. the new section should begin with “Tyrosinase is a Cu-containing…”

Page 2, line 70: UV-Vis is just a part of the electromagnetic spectrum, but not exactly a technique, so please replace “, UV-Vis measurements and…” with “UV-Vis absorption measurements and..” .  Also, it looks like something has gone wrong with the font size in the “Materials and methods” section.

Page 2, line 73: Replace “…thiazole (C1-C3)…” with “…thiazoles (C1-C3)…”

Page 5, line 132: Replace “…presents…” with “…present…”

Table 2. Heading “Inhibitory Mechanizm” should be replaced with “Inhibitory Mechani

Table SI5: Some commas are wrongly shown as “,” rather than “.”

Answer: All comments have been corrected

Reviewer 2 Report

Comments and Suggestions for Authors

The authors reported the synthesis and characterization of three carbazole-based organic compounds. They conducted various investigations, with results clearly presented through tables and figures/schemes. Theoretical calculations were used to support the experimental data. Minor revisions are needed before this manuscript can be published in "Sensors":

- The grammar and spelling throughout the manuscript need to be reviewed. For instance, "inhibition" is misspelled in line 66 (first place), and the whole sentence should be revised; line 132 also needs grammatical correction ("present," not "presents").

- The Table of Contents in the Supplementary Information (SI) document needs correction: Table SI3 is missing, and Table SI2 appears twice.

- The numbering scheme for equations in the Supplementary Information (SI) should be checked. For instance, the number (6) is used for two different equations.

- In Section 3 of the SI document, references should be provided in line 44 to support the statement regarding reported methods. The same applies to Section 4, line 62.

Comments on the Quality of English Language

The grammar and spelling throughout the manuscript need to be reviewed. For instance, "inhibition" is misspelled in line 66 (first place), and the whole sentence should be revised; line 132 also needs grammatical correction ("present," not "presents").

Author Response

REVIEWER #2

The authors reported the synthesis and characterization of three carbazole-based organic compounds. They conducted various investigations, with results clearly presented through tables and figures/schemes. Theoretical calculations were used to support the experimental data. Minor revisions are needed before this manuscript can be published in "Sensors":

  1. The grammar and spelling throughout the manuscript need to be reviewed. For instance, "inhibition" is misspelled in line 66 (first place), and the whole sentence should be revised; line 132 also needs grammatical correction ("present," not "presents").

Answer: Done.

  1. The Table of Contents in the Supplementary Information (SI) document needs correction: Table SI3 is missing, and Table SI2 appears twice.

Answer: Done.

  1. The numbering scheme for equations in the Supplementary Information (SI) should be checked. For instance, the number (6) is used for two different equations.

Answer: Done.

  1. In Section 3 of the SI document, references should be provided in line 44 to support the statement regarding reported methods. The same applies to Section 4, line 62.

Answer: Appropriate citations to SI have been added.

  1. The grammar and spelling throughout the manuscript need to be reviewed. For instance, "inhibition" is misspelled in line 66 (first place), and the whole sentence should be revised; line 132 also needs grammatical correction ("present," not "presents").

Answer: Done.

Reviewer 3 Report

Comments and Suggestions for Authors

In this work, three novel carbazole-based thiazole derivatives were synthesized using different halogen atoms such as chlorine, bromine, and fluorine. Experimental results and quantum chemical simulations are presented and discussed. Furthermore, the molecular docking in the active site of Concanavalin A and the bioavailability for all compounds were evaluated.

The proposed chromophores synthesis, photophysical properties, and bioavailability effect are interesting, and the carbazole-thiazole-based dyes show potential as fluorescent markers. The work can be considered for publication in Sensors after the modifications suggested below:

-        Figure 1. The absorbance and fluorescence spectra should be presented on the Y-axes as Normalized Absorbance and Normalized Fluorescence, respectively (see Figure 3).

-        Table 1. The units of physical quantities should be presented in the Table header.

-        High values ​​for the correlation coefficient (x2) are presented as 1.579 and 1.681. The number of exponentials used to fit fluorescent lifetime measurements could be checked. Typically, a correlation coefficient (x2) value up to 1.2 is well accepted.

-        Figure 2. The black dotted curve should be described in the caption, is it Ludox?

-        The acronym must be defined the first time it is used (see EWG for more details).

-        Table 2. Units and rounding of uncertainties must be presented appropriately (see e.g. 86.35 +/- 7.99  should be  86 +/- 8).

Author Response

In this work, three novel carbazole-based thiazole derivatives were synthesized using different halogen atoms such as chlorine, bromine, and fluorine. Experimental results and quantum chemical simulations are presented and discussed. Furthermore, the molecular docking in the active site of Concanavalin A and the bioavailability for all compounds were evaluated.

The proposed chromophores synthesis, photophysical properties, and bioavailability effect are interesting, and the carbazole-thiazole-based dyes show potential as fluorescent markers. The work can be considered for publication in Sensors after the modifications suggested below:

  1. Figure 1. The absorbance and fluorescence spectra should be presented on the Y-axes as Normalized Absorbance and Normalized Fluorescence, respectively (see Figure 3).

Answer: In Figure 1, the absorption intensity is presented as A/n, while the emission intensity is represented as E/n^5 assuming that the measurement was done linear in wavelength. The electronic absorption and fluorescence spectra were scaled and then normalized to the same value i.e. 1.0 in respect to the long wavelength bands. The scaled intensities versus frequency give a proper comparison for the different molecules. We have changed the X-asis from wavelength to frequency.

  1. Table 1. The units of physical quantities should be presented in the Table header.

Answer: Explanations of symbols used for physicochemical quantities and their units were given in the footnote to the table. Due to the large amount of data, the units are not presented in the table header. For better readability, this data has been moved to the table caption.

  1. High values ​​for the correlation coefficient (x2) are presented as 1.579 and 1.681. The number of exponentials used to fit fluorescent lifetime measurements could be checked. Typically, a correlation coefficient (x2) value up to 1.2 is well accepted.

Answer: The authors agree with the Reviewer's comment that the correlation coefficient of about 1.5-1.7 is a bit too high. The number of exponents used to fit the fluorescence decay curves was checked. For several samples, fluorescence decay curves were re-recorded at a longer time range. In some cases, the high c2 is probably due to the very low fluorescence intensity of the compounds studied.

  1. Figure 2. The black dotted curve should be described in the caption, is it Ludox?

Answer: Yes. Ludox AS-30 colloidal silica 30 wt. % suspension in water was used to determine the Instrument Response Function (IRF). This information is added to the Materials and Methods section in the ESI file.

  1. The acronym must be defined the first time it is used (see EWG for more details).

Answer: Done.

  1. Table 2. Units and rounding of uncertainties must be presented appropriately (see e.g. 86.35 +/- 7.99  should be  86 +/- 8).

Answer: Done.